# Financial incentives to increase stool collection rates for microbiome studies in adult bone marrow transplant patients

Jillian C. Thompson[1], Yi Ren[2], Kristi Romero[1], Meagan Lew[1], Amy T. Bush[1], Julia A. Messina[3], Sin-Ho Jung[2], Sharareh Siamakpour-Reihani[1], Julie Miller[4], Robert R. Jenq[5], Jonathan U. Peled[6], Marcel R. M. van den Brink[6], Nelson J. Chao[1], Mark G. Shrime[7]☯, Anthony D. Sung[1]☯*

1 Division of Hematologic Malignancies and Cellular Therapy, Duke University School of Medicine, Durham, North Carolina, United States of America, 2 Department of Biostatistics and Bioinformatics, Duke University School of Medicine, Durham, North Carolina, United States of America, 3 Division of Infectious Diseases, Duke University School of Medicine, Durham, North Carolina, United States of America, 4 Center for Advanced Hindsight, Duke University School of Medicine, Durham, North Carolina, United States of America, 5 Departments of Genomic Medicine and Stem Cell Transplantation Cellular Therapy, MD Anderson Cancer Center, University of Texas, Houston, Texas, United States of America, 6 Department of Medicine, Adult Bone Marrow Transplant Service, Memorial Sloan Kettering Cancer Center, New York, New York, United States of America, 7 Department of Otolaryngology Head and Neck Surgery, Harvard Medical School, Boston, Massachusetts, United States of America

☯ These authors contributed equally to this work.
* anthony.sung@duke.edu

**Data Availability Statement:** All relevant data are within the paper and its Supporting Information files.

## Abstract

### Introduction

In order to study the role of the microbiome in hematopoietic stem cell transplantation (HCT), researchers collect stool samples from patients at various time points throughout HCT. However, stool collection requires active subject participation and may be limited by patient reluctance to handling stool.

### Methods

We performed a prospective study on the impact of financial incentives on stool collection rates. The intervention group consisted of allogeneic HCT patients from 05/2017-05/2018 who were compensated with a $10 gas gift card for each stool sample. The intervention group was compared to a historical control group of allogeneic HCT patients from 11/2016-05/2017 who provided stool samples before the incentive was implemented. To control for possible changes in collections over time, we also compared a contemporaneous control group of autologous HCT patients from 05/2017-05/2018 with a historical control group of autologous HCT patients from 11/2016-05/2017; neither autologous HCT group was compensated. The collection rate was defined as the number of samples provided divided by the number of time points we attempted to obtain stool.

**Funding:** This study was supported by P30-AG028716-13 Mini#6 (to ADS), R01 HL124112 (to RRJ, ADS), R01-CA203950 (to ADS, NJC), Seres (to ADS), and the ASH Scholar Award (to ADS). The funders had no role in study design, data collection and analysis, decision to publish, or preparation of the manuscript.

**Competing interests:** I have read the journal's policy and the authors of this manuscript have the following competing interests: MRM receives funding from Seres, Evelo, Jazz Pharmaceuticals, Therakos, Amgen, Merck & Co, Inc, Magenta Therapeutics, Smart Immune, Juno, and serves on the DKMS Advisory Board. ADS receives funding from Novartis, Merck, Seres, and serves as a consultant to AVROBIO. There is no overlap between ADS' work with Novartis, Merck, AVROBIO, and this project. While Seres did fund the collection of stool samples, which led to some of the data presented, they had no role in data analysis and interpretation. Other than the competing interest statement, none of these companies will be mentioned by name. This does not alter our adherence to PLOS ONE policies on sharing data and materials.

## Results

There were 35 allogeneic HCT patients in the intervention group, 19 allogeneic HCT patients in the historical control group, 142 autologous HCT patients in the contemporaneous control group (that did not receive a financial incentive), and 75 autologous HCT patients in the historical control group. Allogeneic HCT patients in the intervention group had significantly higher average overall collection rates when compared to the historical control group allogeneic HCT patients (80% vs 37%, p<0.0001). There were no significant differences in overall average collection rates between the autologous HCT patients in the contemporaneous control and historical control groups (36% vs 32%, p = 0.2760).

## Conclusion

Our results demonstrate that a modest incentive can significantly increase collection rates. These results may help to inform the design of future studies involving stool collection.

## Introduction

The human gut microbiome is the myriad of bacteria, archaea, viruses, and fungi that reside in the human gastrointestinal tract [1–3]. In hematopoietic stem cell transplantation (HCT), disruption of the gut microbiome, concomitant of the transplant conditioning regimen, is associated with post-transplant complications such as the development of graft-versus-host disease and infections [4, 5]. Although many strides have been made in investigating the complex relationship between the gut microbiome and its host, further elucidation of the role of the microbiome in patients undergoing HCT is essential in order to improve patient outcomes [1]. The gut microbiome can be studied with next-generation sequencing of microbial nucleic acids that are extracted from human stool samples [2, 6].

Despite knowing how to utilize human stool samples to investigate the microbiome, we have found that the challenge lies in collecting enough stool samples from study participants at various time points throughout the transplant process. Paramsothy et al. found that this challenge exists even when requesting stool samples from healthy donors, demonstrating that approximately 40% of potential donors declined to participate in their study due to the burden of providing stool samples over a six-week period [7]. A different study focused on at-home stool collection, specifically in cancer patients, by Hogue et al. revealed that only 58% of consented patients provided baseline stool samples and only 25% of consented patients provided follow-up stool samples [8]. Unlike other human tissue sampling methods such as drawing blood or swabbing the skin, collecting stool involves more effort on behalf of the patients, especially in the outpatient setting where patients must handle the stool themselves before subsequently placing in a specimen cup [9]. Thus, stool collection compliance in research studies may be hindered as a result of patient apprehension to handling stool due to factors such as embarrassment, disgust, and privacy concerns [10–12]. Furthermore, cancer patients may experience weakness or constipation due to treatment, which can result in noncompliance with stool collection protocols [8].

However, financial incentives may motivate patients to be more willing to provide stool samples, thus leading to increased adherence to study protocols. For example, Green et al. found that both a modest incentive of $10 and a probabilistic incentive of a 10% chance of winning $50 significantly increased rates of another stool-related research activity, fecal

immunochemical testing ($10 incentive 73.3% vs 66.2%, p = 0.04; chance of winning 71.8% vs 66.2%, p = 0.04), despite not increasing colorectal cancer screening via colonoscopy [9, 13]. Incorporating a strategy that includes financial incentives into a research study design can significantly increase the desired outcome [14–16]. Therefore, we believed that we could significantly improve study participant compliance to providing stool samples throughout the HCT process by giving them compensation for their stool samples.

## Materials and methods

This study was approved by the Duke Health Institutional Review Board (IRB), and written informed consent was obtained from all study participants (IRB protocol #Pro00006268 and Pro00078566).

### Defining groups and sample collection

Patients in the intervention group were compensated financially for their stool samples. The intervention group was composed of patients undergoing allogeneic HCT with treatment start dates between 05/11/2017, the date when the financial incentive was implemented, and 05/11/2018. Collection rates, in addition to baseline characteristics, of the allogeneic HCT patients in the intervention group were compared to those of allogeneic HCT patients from a historical control group. The historical control allogeneic HCT patients had treatment start dates between 11/10/2016, the date a study team member started actively managing stool collection in patients through distribution of collection coolers and consistent follow-up, and 05/10/2017. The allogeneic HCT patients in the historical control group were not compensated.

In order to control for potential differences in stool collection over two different time periods, a contemporaneous control group was also included in the study design. The contemporaneous control group consisted of patients undergoing autologous HCT with treatment start dates between 05/11/2017 and 05/11/2018 who were not compensated in any way for their stool samples. Collection rates and baseline characteristics of the autologous HCT patients in the contemporaneous control group were compared to those of autologous HCT patients from a historical control group. The historical control autologous HCT patients had treatment start dates between 11/10/2016 and 05/10/2017, and these patients were not compensated in any way for their stool samples. Of note, no autologous HCT patients, regardless of control group, were compensated in this study; only the allogeneic HCT patients from the intervention group were compensated. Regardless of group, if a patient's HCT treatment start date fell outside of the specified date ranges for a group, this patient was not included in the final analysis in order to prevent overlap between groups.

In this prospective cohort study, allogeneic HCT patients in both the intervention group and the historical control group were required to provide stool samples at the following time points throughout the HCT process: pre-HCT, day 0 (the day of HCT), and days 7, 14, 21, 30, 60, and 90 post-HCT. Since autologous HCT patients do not come to the Adult Blood and Marrow Transplant Clinic as frequently as allogeneic HCT patients, autologous HCT patients in both the contemporaneous control group and the historical control group were only required to provide stool samples at the following time points throughout the HCT process: pre-HCT and days 7, 14, and 90 post-HCT. Fig 1 provides an overview of the study, depicting group comparisons and when samples were collected from each group.

Stool samples were categorized as "inpatient" if scheduled to be provided by the patient while admitted to the hospital at the time of sample collection or "outpatient" if not admitted at the time of sample collection. When samples were collected in the outpatient setting, the patient was provided with a stool collection kit comprised of a stool collection hat, a specimen

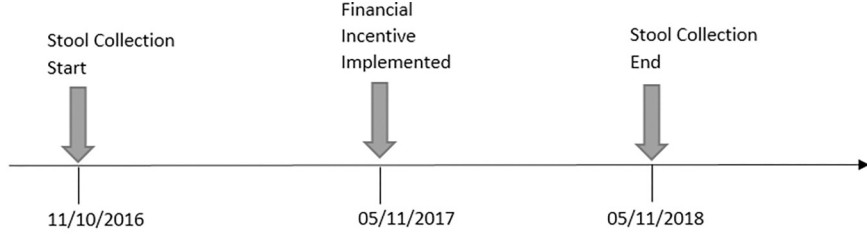

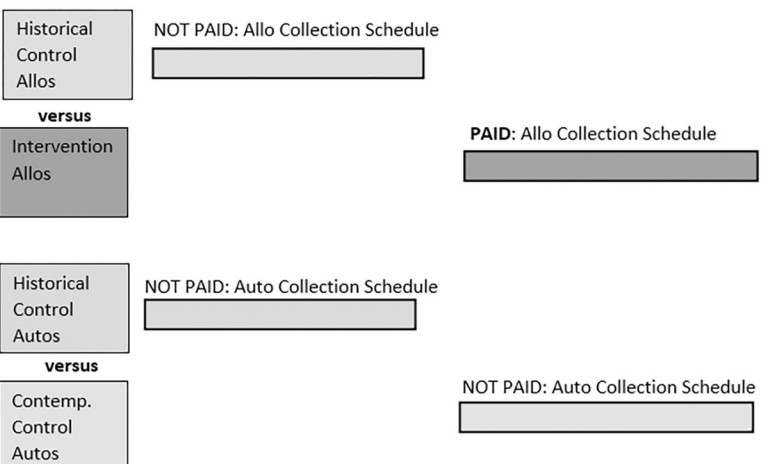

**Fig 1. Sample collection timeline for all groups.** Allo collection schedule: Pre-HCT → Day 0 → Day 7 → Day 14 → Day 21 → Day 30 → Day 60 →Day 90. Auto collection schedule: Pre-HCT → Day 7 → Day 14 → Day 90.

cup, a tongue depressor, and a pair of gloves, along with a cooler and ice pack to store the sample after collecting it themselves. In the inpatient setting, nurses provided patients with a stool collection hat, but the nurses were the ones that performed all the steps of collection and storage after defecation. Patients in the intervention group were allocated a $10 gas gift card for each stool sample provided regardless of whether stool was collected in the inpatient or outpatient setting.

## Data collection

Stool collection was tracked by assessing the number of samples given at their required time points. A collection probability was delineated as the number of samples actually provided by the participant divided by the number of time points for which we required samples be provided. The Duke Adult Blood and Marrow Transplant database was used to query the exact dates that stool samples were collected from each patient in order to verify that samples were provided at the required time points. Only stool samples given between 05/11/2017 and 05/11/2018 were accounted for when determining collection rates for both the intervention and contemporaneous control groups whereas only stool samples given between 11/10/2016 and 05/10/2017 were accounted for when determining collection rates for the historical control group. If a sample was given outside of these time frames, the sample was not included when determining the collection rate. Thus, if a time point typically requiring a sample be given fell outside of these time frames, that time point was not included when assessing compliance, neither hurting the participant's collection rate if no sample was given, nor helping the participant if a sample was given. Furthermore, if a participant withdrew from the study or died, then the

subsequent time points after date of death or withdrawal were not included in the analysis. Each sample was tracked for whether it was provided in the inpatient or outpatient setting in order to assess inpatient and outpatient collection rates. Demographic data such as age, gender, race, ethnicity, disease, and conditioning type were abstracted from the Duke Adult Blood and Marrow Transplant database and from electronic medical records.

### Statistical analysis

Baseline demographics were summarized with N (%) for categorical variables and median (interquartile range) with mean (standard deviation) for continuous variables for all patients. Chi-square tests or Fisher's exact tests were utilized to compare categorical variables, as appropriate, and Wilcoxon Rank Sum tests or t-tests were utilized to compare continuous variables, as appropriate. For allogeneic patients, negative binomial regression with generalized estimating equation (GEE) was performed to model the inpatient and outpatient collection rates of each patient, if applicable. GEE with compound symmetry correlation structure was used to account for the correlation of the two rates for each patient. Other covariates such as age, gender, race, disease, and conditioning type were adjusted for in order to avoid confounding. All analyses were conducted using SAS version 9.4 (SAS Institute, Cary, NC) and R version 3.5.0.

A priori sample size calculation was not performed for this study. While this study was approved by the Duke Health IRB and written informed consent was obtained from all study participants, this study was not formally planned.

### Results

Fifty-four patients undergoing allogeneic HCT and 217 patients undergoing autologous HCT were included in the study cohort. Of the 54 allogeneic HCT patients, there were 35 (64.8%) allogeneic HCT patients in the intervention group that were compared to 19 (35.2%) allogeneic HCT patients in the historical control group. Although not significantly different, the intervention group tended to be slightly older at transplant (61 vs 51 median age, p = 0.0853) and included a smaller proportion of female patients (28.6% vs 52.6%, p = 0.0804). There were also no significant differences between the two groups of allogeneic HCT patients with regard to other baseline demographics such as race, ethnicity, disease, and conditioning (Table 1).

Of the 217 autologous HCT patients, there were 142 (65.3%) autologous HCT in the contemporaneous control group that were compared to 75 (34.7%) autologous HCT patients in the historical control group. The majority of patients in both groups received autologous HCT to treat multiple myeloma. There were no significant differences between the two groups of autologous HCT patients with regard to baseline demographics such as age at transplant, gender, race, ethnicity, and disease (Table 2).

The allogeneic HCT patients in the intervention group displayed better compliance to stool collection protocols than the allogeneic HCT patients in the historical control group (Table 3). For instance, the mean overall collection rate in the intervention group of allogeneic HCT patients was much higher than the mean overall collection rate of the allogeneic HCT patients in the historical control group (80% vs 37%, p<0.0001). In addition to an increased mean overall collection rate, the allogeneic HCT patients in the intervention group also demonstrated significantly increased mean outpatient collection rates (84% vs 23%, p<0.0001) and significantly increased mean inpatient collection rates (71% vs 46%, p = 0.0409).

On the other hand, there were no differences in compliance to stool collection protocols between the autologous patients in the contemporaneous control and historical control groups (Table 4). Mean overall collection rates were similar in both groups of autologous patients (36% vs 32%, p = 0.2760). Furthermore, mean outpatient collection rates (30% vs 28%,

**Table 1. Baseline allogeneic HCT patient characteristics.**

| | Intervention Group | Historical Control Group | All Patients | |
|---|---|---|---|---|
| | N = 35 (64.8%) | N = 19 (35.2%) | N = 54 (100%) | P-Value |
| Age at Transplant, median (IQR)* | 61 (50–64) | 51 (35–59) | 56 (46–63) | 0.0853 |
| Gender, female, no. (%)** | 10 (28.6%) | 10 (52.6%) | 20 (37%) | 0.0804 |
| Race, no. (%) | | | | |
| Black/African American | 2 (5.7%) | 4 (21.1%) | 6 (11.1%) | 0.2693 |
| Other/Unknown | 2 (5.7%) | 0 (0%) | 2 (3.7%) | |
| White | 31 (88.6%) | 15 (78.9%) | 46 (85.2%) | |
| Ethnicity, no. (%) | | | | |
| Hispanic or Latino | 1 (2.9%) | 1 (5.3%) | 2 (3.7%) | 1.0000 |
| Not Hispanic or Latino | 33 (94.3%) | 18 (94.7%) | 51 (94.4%) | |
| Unknown | 1 (2.9%) | 0 (0%) | 1 (1.9%) | |
| Disease, no. (%) | | | | |
| Acute Leukemia | 13 (37.1%) | 8 (42.1%) | 21 (38.9%) | 0.4396 |
| Lymphoma | 4 (11.4%) | 4 (21.1%) | 8 (14.8%) | |
| MDS/MPN | 14 (40%) | 4 (21.1%) | 18 (33.3%) | |
| Multiple Myeloma | 1 (2.9%) | 2 (10.5%) | 3 (5.6%) | |
| Other | 3 (8.6%) | 1 (5.3%) | 4 (7.4%) | |
| Myeloablative Conditioning, no. (%) | 23 (65.7%) | 13 (68.4%) | 36 (66.7%) | 0.8403 |

*t-test was used to test age difference and Wilcoxon Rank Sum tests were used for other continuous variables.

**Chi-squared test was used to test gender difference and Fisher's exact tests were used for other categorical variables.

p = 0.5360) and mean inpatient collection rates (46% vs 59%, p = 0.2509) were comparable as well. Fig 2A demonstrates the proportion of stool samples collected at each time point in the outpatient setting, whereas Fig 2B demonstrates the proportion of stool samples collected in

**Table 2. Baseline autologous HCT patient characteristics.**

| | Contemporaneous Control Group | Historical Control Group | All Patients | |
|---|---|---|---|---|
| | N = 142 (65.3%) | N = 75 (34.7%) | 217 (100%) | P-Value |
| Age at Transplant, median (IQR)* | 60 (53–67) | 62 (53–67) | 61 (53–67) | 0.6255 |
| Gender, female, no. (%)** | 55 (38.7%) | 35 (46.7%) | 90 (41.5%) | 0.2592 |
| Race, no. (%) | | | | |
| Black/African American | 31 (21.8%) | 19 (25.3%) | 50 (23%) | 0.6697 |
| Other/Unknown | 7 (4.9%) | 5 (6.7%) | 12 (5.5%) | |
| White | 104 (73.2%) | 51 (68%) | 155 (71.4%) | |
| Ethnicity, no. (%) | | | | |
| Hispanic or Latino | 3 (2.1%) | 2 (2.7%) | 5 (2.3%) | 0.2176 |
| Not Hispanic or Latino | 138 (97.2%) | 70 (93.3%) | 208 (95.9%) | |
| Unknown | 1 (0.7%) | 3 (4%) | 4 (1.8%) | |
| Disease, no. (%) | | | | |
| Acute Leukemia | 1 (0.7%) | 0 (0%) | 1 (0.5%) | 0.9319 |
| Lymphoma | 39 (27.5%) | 18 (24%) | 57 (26.3%) | |
| Multiple Myeloma | 96 (67.6%) | 54 (72%) | 150 (69.1%) | |
| Other | 6 (4.2%) | 3 (4%) | 9 (4.1%) | |

*t-test was used to test age difference and Wilcoxon Rank Sum tests were used for other continuous variables.

**Chi-squared test was used to test gender difference and Fisher's exact tests were used for other categorical variables.

**Table 3. Allogeneic HCT patient stool collection rates.**

| | Intervention Group | Historical Control Group | All Patients | |
|---|---|---|---|---|
| | N = 35 (64.8%) | N = 19 (35.2%) | N = 54 (100%) | P-Value |
| Overall Collection Rate | | | | |
| Median (IQR) | 0.875 (0.75–1) | 0.375 (0–0.67) | 0.75 (0.375–0.875) | < .0001 |
| Mean (SD) | 0.80 (0.24) | 0.37 (0.36) | 0.65 (0.35) | |
| Outpatient Collection Rate | | | | |
| Median (IQR) | 1 (0.8–1) | 0 (0–0.5) | 0.82 (0.25–1) | < .0001 |
| Mean (SD) | 0.84 (0.27) | 0.23 (0.33) | 0.64 (0.41) | |
| Inpatient Collection Rate | | | | |
| Median (IQR) | 0.8 (0.5–1) | 0.5 (0–0.75) | 0.75 (0.4–1) | 0.0409 |
| Mean (SD) | 0.71 (0.36) | 0.46 (0.41) | 0.62 (0.40) | |

*Wilcoxon Rank Sum tests were used to test the rate differences.

the inpatient setting, amongst the allogeneic and autologous transplant patients in the intervention and control groups.

Table 5 displays a multivariate analysis modeling sample collection rates amongst the allogeneic transplant patients. The stool sample collection rate was 3.853 times higher in the intervention group than the stool sample collection rate in the historical control group (95% CI: 1.938, 7.657). There were no overall significant differences in sample collection rates after adjusting for covariates such as age, gender, conditioning, race, and disease. However, allogeneic transplant patients with lymphoma, MDS/MPN, or multiple myeloma, had significantly higher incidence rate ratios for sample collection rates when compared to allogeneic transplant patients with acute leukemia. Furthermore, African American allogeneic transplant patients had 2.658 times higher stool sample collection rates when compared to white allogeneic transplant patients (95% CI: 1.36, 5.194).

## Discussion

With a significant increase in overall, outpatient, and inpatient collection rates in the intervention group, our results indicate that even moderate incentivization in the form of a $10 gas gift card can be efficacious in improving stool collection compliance in research. While this stands in contrast to other studies of $5-$20 incentives that showed no increase in collection rates of at-home fecal immunochemical testing or fecal occult blood testing, we believe our study

**Table 4. Autologous HCT patient stool collection rates.**

| | Contemporaneous Control Group | Historical Control Group | All Patients | |
|---|---|---|---|---|
| | N = 142 (65.3%) | N = 75 (34.7%) | 217 (100%) | P-Value |
| Overall Collection Rate | | | | |
| Median (IQR) | 0.25 (0–0.75) | 0.25 (0–0.5) | 0.25 (0–0.67) | 0.2760 |
| Mean(SD) | 0.36 (0.35) | 0.32 (0.37) | 0.35 (0.36) | |
| Outpatient Collection Rate | | | | |
| Median (IQR) | 0 (0–0.67) | 0 (0–0.5) | 0 (0–0.67) | 0.5360 |
| Mean(SD) | 0.30 (0.36) | 0.28 (0.38) | 0.29 (0.37) | |
| Inpatient Collection Rate | | | | |
| Median (IQR) | 0.5 (0–1) | 1 (0–1) | 0.5 (0–1) | 0.2509 |
| Mean(SD) | 0.46 (0.47) | 0.59 (0.50) | 0.49 (0.47) | |

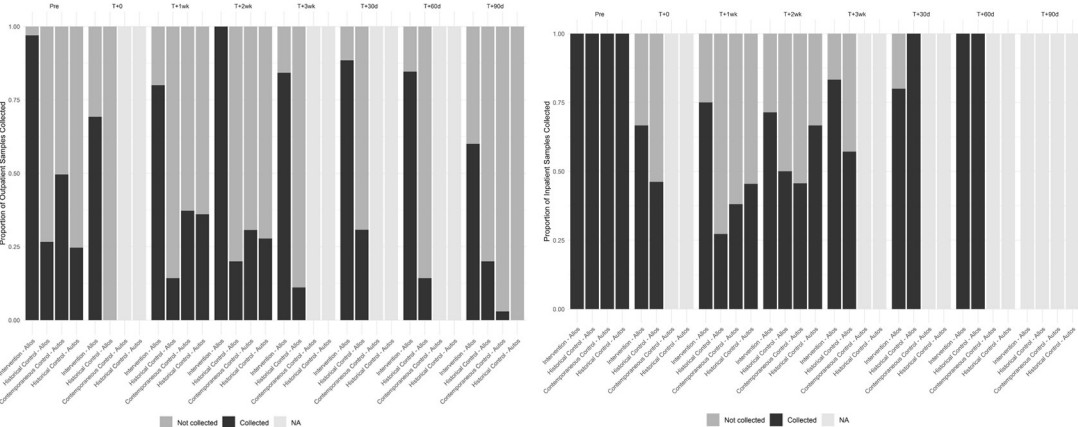

**Fig 2. (a) Outpatient Collections across Time Points for All Groups.** Each collection time point is indicated at the top of the figure: Pre, T+0 (Day 0), T+ 1wk (Day 7), T+ 2wk (Day 14), T+ 3wk (Day 21), T+ 30d (Day 30), T+ 60d (Day 60), T+ 90d (Day 90). At each time point, the proportion of samples collected/not collected are shown for each group. If denoted as 'collected' (represented in black), this proportion of samples was successfully provided. If denoted as 'not collected' (represented in dark gray), this proportion of samples was not provided. If denoted as 'NA' (represented in light gray), this time point was not a required collection time point for that particular group. **(b) Inpatient Collections across Time Points for All Groups.** Each collection time point is indicated at the top of the figure: Pre, T+0 (Day 0), T+ 1wk (Day 7), T+ 2wk (Day 14), T+ 3wk (Day 21), T+ 30d (Day 30), T+ 60d (Day 60), T+ 90d (Day 90). At each time point, the proportion of samples collected/not collected are shown for each group. If denoted as 'collected' (represented in black), this proportion of samples was successfully provided. If denoted as 'not collected' (represented in dark gray), this proportion of samples was not provided. If denoted as 'NA' (represented in light gray), this time point was not a required collection time point for that particular group or, in the case of the day 90 time point, none of the samples were provided in the inpatient setting at this time point.

**Table 5. Negative binomial regression with GEE on stool sample collection rate of allogeneic transplant patients.**

| | Incident Rate Ratio (95% CI) | P-Value | Overall P-Value |
|---|---|---|---|
| Group | | | |
| Historical Control Group | -REF- | | 0.001 |
| Intervention Group | 3.853 (1.938–7.657) | <0.001 | |
| Age | | | |
| Continuous | 0.987 (0.971–1.003) | 0.112 | |
| Gender | | | |
| Male | -REF- | | 0.365 |
| Female | 0.756 (0.431–1.328) | 0.331 | |
| Conditioning | | | |
| Myeloablative | -REF- | | 0.140 |
| NMA/RIC | 0.603 (0.321–1.132) | 0.115 | |
| Race | | | |
| White | -REF- | | 0.098 |
| Black/African American | 2.658 (1.36–5.194) | 0.004 | |
| Other/Unknown | 0.967 (0.398–2.35) | 0.942 | |
| Disease | | | |
| Acute Leukemia | -REF- | | 0.068 |
| Lymphoma | 2.345 (0.938–5.863) | 0.068 | |
| MDS/MPN | 1.84 (1.039–3.258) | 0.036 | |
| Multiple Myeloma | 5.146 (1.993–13.287) | <0.001 | |
| Other | 0.625 (0.183–2.134) | 0.454 | |

demonstrates that a modest financial incentive of $10 for each stool sample is effective in procuring higher rates of stool samples for a couple of possible reasons [17–19]. For instance, the serial collection design of the study, requiring stool samples at multiple time points. The multiple time points provide a study participant in the intervention group multiple opportunities to earn a $10 gas gift card for each stool sample, thus the potential to actually earn more than $10 in gas gift cards during the entirety of the study. Another possible contributing factor for the effectiveness of financial incentives in our study was that study participants had the opportunity to return their required stool samples in-person at their clinic appointments, avoiding having to mail the sample which may be perceived by some as an additional inconvenience. Furthermore, employing a contemporaneous control group that did not receive the financial incentive into the study design addresses the possible confounders associated with potential discrepancies in stool collection rates over time, strengthening our finding that the increase in collection rates can be attributed to the financial incentive.

Despite the effectiveness of the financial incentive, our study is not without limitations. For instance, although accounted for in the statistical analyses, there are considerable differences in sample size between not only the comparison groups within each transplant type, but also between the total number of allogeneic and autologous transplant patients included in the study. The difference in the number of allogeneic and autologous transplant study participants is reflective of our patient population: about twice as many adult autologous stem cell transplants are performed each year than adult allogeneic transplants at Duke. Another limitation of the study is that the financial incentive was only made available to allogeneic transplant patients due to funding restraints; this was accounted for by only performing comparisons within the same transplant type. The non-randomization of the study is also a limiting factor because it does not take into account confounders such as social determinants of health that may make someone more or less inclined to participate in a research study involving financial incentives. Furthermore, although it was found that African American allogeneic transplant patients had higher stool sample collection rates when compared to white allogeneic transplant patients, there is a lack of racial and ethnic diversity in this study with the majority of study participants being non-Hispanic whites.

Effort on behalf of the patient is required most when providing a stool sample in the outpatient setting since patients must do the collection process themselves, as opposed to the inpatient setting where nursing staff aid with stool collection for admitted patients. Thus, the formidable boost in collection rates in the outpatient setting in the intervention group underscore the role of the financial incentive in this study. While the increase in inpatient collection rates in the intervention group is still significant, the average inpatient collection rate associated with the intervention group is mediated in the part by the role of nurses who work with patients to collect samples in that setting. Also, inpatient collection time points may have been missed when patients were only admitted for 24–48 hours for indications such as febrile neutropenia before being discharged to continue antibiotics in the outpatient setting, thus leaving a very narrow window for inpatient collection.

Furthermore, when utilizing financial incentives to motivate desired behaviors from patients, it is imperative to ensure the use of the financial incentive is ethical. For example, Bartholomew et al. raised concerns over the use of financial incentives in breast cancer screening, stating they may stimulate patients to make decisions they would not otherwise make, which can subsequently lead to patient harm (e.g. unnecessary treatment) [20]. However, the use of financial incentives in the context of our study is quite different. A participant's decision to provide stool does not affect their treatment plan nor do they receive any sequencing results from their stool samples. The samples are not used for any sort of screening and have no effect on participant health; the stool samples are solely used for research purposes in order to

improve outcomes of future patients. Undue inducement is another ethical concern that arises with financial compensation to research participants [21]. However, we believe the amount of $10 per stool sample was modest enough to not unduly influence a patient's decision to participate in the study while still serving as an effective incentive to improve patient compliance to stool collection protocols.

While this study was performed in a specialized HCT patient population, this study design utilizing financial incentives to increase stool collection rates may be able to be executed in a myriad of patient populations. If these results are generalizable, other researchers attempting to procure stool samples for microbiome studies may be able to increase their patient compliance and improve their stool collection rates. Future directions for this study will be to observe the use of financial incentives for stool collection in the HCT population longitudinally in order to evaluate whether the effectiveness of the financial incentive would wear off over time. Furthermore, with more funding, autologous HCT patients can be included in the study. Another next step is to investigate how social determinants of health affect stool collection rates in the HCT population, identifying how factors such as socioeconomic status influence compliance and willingness to participate in a study utilizing financial incentives.

## Supporting information

**S1 Data.**
(XLSX)

**S2 Data.**
(XLSX)

## Author Contributions

**Conceptualization:** Robert R. Jenq, Marcel R. M. van den Brink, Nelson J. Chao, Mark G. Shrime, Anthony D. Sung.

**Data curation:** Jillian C. Thompson, Yi Ren, Kristi Romero, Meagan Lew, Amy T. Bush, Anthony D. Sung.

**Formal analysis:** Jillian C. Thompson, Yi Ren, Kristi Romero, Meagan Lew, Sin-Ho Jung, Mark G. Shrime, Anthony D. Sung.

**Funding acquisition:** Robert R. Jenq, Marcel R. M. van den Brink, Nelson J. Chao, Anthony D. Sung.

**Investigation:** Jillian C. Thompson, Yi Ren, Kristi Romero, Meagan Lew, Amy T. Bush, Julia A. Messina, Jonathan U. Peled, Marcel R. M. van den Brink, Nelson J. Chao, Mark G. Shrime, Anthony D. Sung.

**Methodology:** Jillian C. Thompson, Yi Ren, Kristi Romero, Meagan Lew, Amy T. Bush, Julia A. Messina, Robert R. Jenq, Nelson J. Chao, Mark G. Shrime, Anthony D. Sung.

**Project administration:** Jillian C. Thompson, Kristi Romero, Meagan Lew, Amy T. Bush, Sharareh Siamakpour-Reihani, Nelson J. Chao, Anthony D. Sung.

**Resources:** Sharareh Siamakpour-Reihani, Julie Miller, Nelson J. Chao, Anthony D. Sung.

**Supervision:** Kristi Romero, Julia A. Messina, Sharareh Siamakpour-Reihani, Nelson J. Chao, Mark G. Shrime, Anthony D. Sung.

**Validation:** Jillian C. Thompson, Kristi Romero, Amy T. Bush, Nelson J. Chao, Mark G. Shrime, Anthony D. Sung.

**Writing – original draft:** Jillian C. Thompson, Yi Ren, Mark G. Shrime, Anthony D. Sung.

**Writing – review & editing:** Jillian C. Thompson, Yi Ren, Kristi Romero, Meagan Lew, Amy T. Bush, Julia A. Messina, Sin-Ho Jung, Sharareh Siamakpour-Reihani, Julie Miller, Robert R. Jenq, Jonathan U. Peled, Marcel R. M. van den Brink, Nelson J. Chao, Mark G. Shrime, Anthony D. Sung.

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
