## [Decision Letter · Decision Letter 0]

15 Feb 2022

PONE-D-21-40817Financial incentives to increase stool collection rates for microbiome studies in adult bone marrow transplant patientsPLOS ONE

Dear Dr. Sung,

Thank you for submitting your manuscript to PLOS ONE. After careful consideration, we feel that it has merit but does not fully meet PLOS ONE’s publication criteria as it currently stands. Therefore, we invite you to submit a revised version of the manuscript that addresses the points raised during the review process.

 Please provide detailed responses to all comments from the reviewers, especially regarding the sample size of the study and the methodology involving the calculation of primary endpoint.

We look forward to receiving your revised manuscript.

Kind regards,

Peter Gyarmati

Academic Editor

PLOS ONE

Journal Requirements:

2. Thank you for stating the following in the Competing Interest section: 

"I have read the journal's policy and the authors of this manuscript have the following competing interests:

MRM receives funding from Seres, Evelo, Jazz Pharmaceuticals, Therakos, Amgen, Merck & Co, Inc, Magenta Therapeutics, Smart Immune, Juno, and serves on the DKMS Advisory Board. ADS receives funding from Novartis, Merck, Seres, and serves as a consultant to AVROBIO. There is no overlap between ADS’ work with Novartis, Merck, AVROBIO, and this project. While Seres did fund the collection of stool samples, which led to some of the data presented, they had no role in data analysis and interpretation. Other than the competing interests statement, none of these companies will be mentioned by name."

We note that you received funding from a commercial source: Seres, Evelo, Jazz Pharmaceuticals, Therakos, Amgen, Merck & Co, Inc, Magenta Therapeutics, Smart Immune, Juno,Novartis, Merck and Seres. 

Reviewers' comments:

Reviewer's Responses to Questions

**Comments to the Author**

1. Is the manuscript technically sound, and do the data support the conclusions?

Reviewer #1: No

Reviewer #2: Yes

2. Has the statistical analysis been performed appropriately and rigorously? 

Reviewer #1: No

Reviewer #2: Yes

3. Have the authors made all data underlying the findings in their manuscript fully available?

Reviewer #1: Yes

Reviewer #2: Yes

4. Is the manuscript presented in an intelligible fashion and written in standard English?

Reviewer #1: Yes

Reviewer #2: Yes

5. Review Comments to the Author

Reviewer #1: The authors undertook a controlled before after study to measure the effect of a $10 petrol gift card on the collection of stool samples for patients undergoing allogenic hematopoietic stem cell transplantation. I have some concerns about the research reported here, both technical and ethical.

please see the file attached for detailed comments.

Reviewer #2: This timely article addresses a very hot and important topic of compensation in human microbiome research. The authors performed a prospective study on the impact of financial incentives on stool collections rates among stem cell transplantation patients and demonstrated that a modest incentive of 10 dollars can significantly increase collection rates. This results may inform the design of future studies involving stool collection, given the rapid development of human microbiome research, the value of study is obvious. However, there are still some weaknesses of this paper.

Firstly, the author may need to explain why $10？what's the normal compensation amount in clinical trials setting in their country?

Secondly, the authors may need to consider the appropriateness of the compensation type regarding the participant population, what about other types of compensation?

Thirdly, Financial compensation is closely related with undue inducement, which refers to compensation that is so great in amount or nature that it decreases participants' ability to rationally consider participation in the study. The compensation should not be so large that individuals may feel compelled to participate. The authors are suggested to include a discussion on how to avoid undue inducement while incentivise participation and stool collection compliance in research.

Fourthly, is there any particular vulnerabilities with the patient group the paper focused? would their vulnerabilities affect their participation in fecal collection?

If authors want to get back to the work by addressing these issues, this would make for a far more targeted and clearer discussion.

6. PLOS authors have the option to publish the peer review history of their article (what does this mean?). If published, this will include your full peer review and any attached files.

Reviewer #1: **Yes: **Isabelle Durand-Zaleski

Reviewer #2: **Yes: **Yonghui Ma

---

## [Author Response · Author response to Decision Letter 0]

31 Mar 2022

Response to Reviewers 

Thank you very much for consideration of our manuscript, and we greatly appreciate your feedback and commitment to the peer review process. Please see a point-by-point response to the reviewers’ comments. We have revised the paper accordingly.

Reviewer #1: The authors undertook a controlled before after study to measure the effect of a $10 petrol gift card on the collection of stool samples for patients undergoing allogenic hematopoietic stem cell transplantation. I have some concerns about the research reported here, both technical and ethical.

please see the file attached for detailed comments.

The topic of financial incentives for patients is debated on grounds of both efficacy and ethics. A checklist was developed for healthcare professionals (Glasziou PP, Buchan H, Del Mar C, Doust J, Harris M, Knight R, Scott A, Scott IA, Stockwell A. When financial incentives do more good than harm: a checklist. BMJ. 2012 Aug 13;345:e5047. doi: 10.1136/bmj.e5047. PMID: 22893568.) and some of the items can also be relevant for financial incentives directed to patients: 

Thank you for suggesting this helpful framework for assessing the appropriateness of a financial incentive. After thorough consideration of the checklist, we believe that the financial incentive utilized in our study is appropriate, as we can answer ‘yes’ to the aforementioned questions. Please see below for responses to the specific questions: 

1. Does the desired action improve patient outcomes?

Response: Yes, we believe the desired action of obtaining stool samples from patients undergoing hematopoietic stem cell transplantation (HCT) will improve patient outcomes by providing invaluable information about the role of the microbiome in HCT patients, as discussed in the introduction of our manuscript. In addition to the introduction of our manuscript, a recently published paper by some of our co-authors in The New England Journal of Medicine further emphasized the pertinence of learning more about the microbiome of HCT patients by demonstrating that lower mortality was associated with increased diversity of intestinal microbiota at the time of engraftment in patients undergoing allogeneic HCT, an association gleaned from the analysis of many stool samples (van den Brink MD et al. Microbiota as Predictor of Mortality in Allogeneic Hematopoietic-Cell Transplantation. N Engl J Med. 2020 Feb 27; 382:822-834. doi: 10.1056/NEJMoa1900623. PMID: 32101664). Learning more about the microbiome in HCT patients, and most importantly learning more about how to utilize this data to decrease morbidity and mortality, necessitates collecting stool from these patients.

 2. Will undesirable behaviour persist without intervention?

Response: Yes, we believe that undesirable behavior (patient noncompliance with stool collection protocols) will persists without intervention (modest, noncoercive financial incentive), as demonstrated by the results of our manuscript. For instance, allogeneic HCT patients in the intervention group had significantly higher average overall collection rates when compared to the historical control group allogeneic HCT patients (80% vs 37%, p<0.0001). Collection rates were also low in the autologous HCT patients in the contemporaneous control and historical control groups who were not compensated with no significant difference in the collection rates between these groups of autologous patients (36% vs 32%, p=0.2760). In addition to the results of our manuscript, other studies also demonstrated difficulty in trying to obtain stool samples from patients (please see introduction of this manuscript for specific studies, line 77-90). Thus, we believe that poor stool sample collection rates will persist without the utilization of modest financial incentives.

3. Are there valid, reliable, and practical measures of the desired behaviour?

Response: Yes, there are valid, reliable, and practical measures of stool collection. Assessing stool collection through collection rate, defined as the number of samples provided divided by the number of time points we attempted to obtain stool, is an objective way to measure participant compliance to providing stool samples. We also believe that assessing stool collection this way is reliable (precise and reproducible) and will be able to be reproduced by other teams attempting to collect stool for microbiome studies.

4. Have the barriers and enablers to improving behaviour been assessed?

Response: Yes, the barriers and enablers to improving patient compliance with providing stool samples was assessed through literature review. Please see introduction of manuscript (lines 77-90).

5. Will financial incentives work, and better than other interventions to change behaviour, and why?

Response: Yes, we believe that financial incentives will work in improving patient compliance in providing stool samples, based on the results of our studies and others mentioned in the introduction of the manuscript. We also believe that financial incentives work better than other interventions based on the stool collection rates of the control groups. For example, all patients, regardless of group, receive education on how to provide stool by the study coordinator during the consent process, as well as reminders to bring their stool samples with them at required time points. While not specifically assessed, the low compliance in the groups who were not compensated demonstrate that other methods to improving compliance such as patient education and reminders are not enough to significantly improve stool collection rates. This was not specifically assessed, but can be inferred by the results of the study. 

6. Will benefits clearly outweigh any unintended harmful effects, and at an acceptable cost?

Response: Yes, the benefits to collecting stool (please see response to question 1) outweigh any unintended harmful effects. Furthermore, our IRB-approved protocols (Pro00006268, Pro000078566) lists no “Expected Adverse Events” for stool collection: ‘

Stool Collection: There are no known risks associated with the collection of stool samples. There may be risks, discomforts, or side effects that are not yet known.” 

Also, providing stool as part of the study does not affect the patient-clinician relationship since a study coordinator not involved in the direct care of the patient handles consenting and other study-related tasks. 

The authors undertook a controlled before after study to measure the effect of a $10 petrol gift card on the collection of stool samples for patients undergoing allogenic hematopoietic stem cell transplantation. I have some concerns about the research reported here, both technical and ethical. 

General comments

I think a discussion on the appropriateness and ethics of the financial incentive in this context, based for example on the BMJ checklist and on the following article (Bartholomew T, Colleoni M, Schmidt H. Financial incentives for breast cancer screening undermine informed choice. BMJ. 2022 Jan 10;376:e065726. doi: 10.1136/bmj-2021-065726. PMID: 35012959.), would be welcome. Also I would question the opportunity of gift cards for petrol at a time when we are globally trying to reduce pollution. Why the choice of petrol and not vouchers for cultural goods as in BMJ 2021;375:e065217 for example? 

Response: Thank you for this recommendation. Bartholomew et al. make some excellent points in regards to the role of financial incentives in breast cancer screening. However, we feel that the use of financial incentives to increase stool collection in the context of our study is quite different for the following reasons. For example, unlike breast cancer screening that may lead to psychological distress from false positives or unnecessary treatment as mentioned in the article, participants providing stool samples in the context of our study does not affect their treatment plan nor do they receive any sequencing results from their stool samples. The samples are not used for any sort of screening and have no effect on participant health. The process of providing stool is noninvasive, and there are no known risks associated with the collection of stool samples. The stool samples are solely used for research purposes in order to learn more about how to use the microbiome to improve outcomes of future patients. Furthermore, unlike the process of consenting for breast cancer screening by physicians attempting to make screening quotas, there is no undue influence driven by conflicts of interest during the consenting process for our study since a study coordinator who is not a part of the healthcare team does the consenting. Finally, the Bartholomew et al. article mentions that “the fundamental ethical concern with incentives is that they may lead people to make choices they would not have made without the incentive are harmed by this choice.” There is no known harm involved in providing a stool sample since it has no influence on patient care as described above, and we used an IRB-approved study protocol that deemed $10 modest enough to not have undue influence on patient decision making. The consent forms were also part of this IRB approved study and were thoroughly explained to patients by study coordinators with a sole focus of promoting patient autonomy and well-being. Thus, we believe the use of financial incentives in our study to be ethical. 

Please also note the stool and blood samples are collected based on our standing IRB approved biorepository protocols (Pro00006268, Pro000078566). Participation is voluntarily and will not affect the treatment as required by the IRB. The stool collection protocol was not stablished around the $10 incentives. 

In regards to the decision to use petrol gift cards as the incentive, we agree that it is very important to be environmentally conscious; however, many of our patients are driving very long distances to reach their appointments at the Duke Adult Bone Marrow Transplant Clinic. The majority of our patients are not local patients. They might have been driving from rural areas of NC or even other states. These patients have no other option but to use gas in order to get to their appointments. Thus, gas gift cards for $10 was an option that we felt would be an effective incentive without too much value to be considered coercive. 

Furthermore, due to the nature of the disease, difficult treatment and post HCT periods of time, other vouchers might not be appropriate or useful to our patient population. The mortality and morbidity rate of HCT post treatment can be as high as 30%, these are venerable and immune suppressed patient population. Even without a global pandemic, many of the cultural events or social settings won’t be appropriate in this setting. 

Hopefully by 2030-2035 (EU and US), when the goal is to have most of the cars independent of gas, our patients have the option not to drive traditional cars. Thus, help with environmental protection. As of now electric cars are out of reach of majority of the world population for many reasons including cost.

Do patients derive a personal benefit from the microbiome study, in terms of treatment adaptation for example, or is it only for research that will eventually benefit other patients later?

Response: That’s a great question. No, patients do not derive a personal benefit from the microbiome study. These samples are just used for research that is aimed to benefit future patients. 

For the primary endpoint, I would like an explanation of how it was calculated. What does a mean rate of 80% mean in Table 3, is it 80% of the population who brought the planned 8 stools? Or is it the total number of stools divided by 8 and by the total population? What is the most important for you research, to have the required number of samples for the population of to have as many patients as possible with at least one stool?

Response: It means the average of all the patients’ collection rates. Collection rate is defined as the number of submitted samples divided by the number of required or maximum possible number of submissions. Collection rate is more important for our research, as opposed to as many patients as possible with at least one stool, in order to assess the changes to the microbiome over time as a patient goes through the HCT process. In addition to that, our objective was trying to get more samples from one patient, instead of just one sample from one patient. 

Specific comments:

Why is collection rate, and not the total number of stools collected chosen as the endpoint? 

Response: Please see answer to previous question above. Furthermore, the allogeneic and autologous transplant patients have different sample collection timelines since allogeneic patients have to come to the clinic more frequently after transplant due to the more rigorous post-transplant recovery. Thus, we attempted to collect stool samples from the allogeneic patients at the following time points: pre-HCT, Day 0, Day 7, Day 14, Day 21, Day 30, Day 60, and Day 90. On the other hand, stool samples were only collected from autologous patients at pre-HCT, Day 7, Day 14, and Day 90. Plus, if a participant withdrew from the study or died, then the subsequent time points after date or death or withdrawal were not included in the analysis. Thus, the maximum number of possible samples collected from each patient can be different.

Also I am not sure that the term ‘rate’ is mathematically correct if you mean percentage. Is time a variable included in your rate calculations? 

Response: Time is not a variable included in the rate calculations. In this context, we believe ‘rate’ is the correct calculation as it presents a frequency of discrete submissions.

I understood from Fig 1 that 8 samples are planned. Why not report the average number of stools collected per patient and per period (before/after)? 

Response: We intended for Figure 1 to represent an overview of the sample collection timeline for the different groups. Stool samples were planned to be collected from allogeneic HCT patients at 8 different time points whereas samples were planned to be collected from autologous HCT patients at 4 different time points. The group of historical control allogeneic patients consists of different patients than the intervention allogeneic patients. That is why we didn’t compare before and after. The historical control autologous group consist of different patients than the contemporaneous control autologous group. Please see “Defining groups and sample collection” of the Materials and methods section. 

Why is the total number of stools collected not reported in the auto group either? I saw that all the figures are I the excel file so it would be possible to present at least the endpoints that I think would be relevant:

- % of patients with a full stool collection before/ after and in the control group.

- % of the total number of samples collected (N collected / N population x8)

Response: We chose not to report the absolute number since the collection rate better portrays compliance to stool collection protocol since the allogeneic and autologous patients have different sample collection schedule and to account for patient death and dropout. Please see above. 

Why is there no sample size calculation? 

Response: A priori sample size calculation was not performed for this study. While this study was approved by the Duke IRB and written informed consent was obtained from all study participants, this study was not formally planned. However, all statistical analyses were performed appropriately, and the limitations related to the sample size in our study have been addressed in the discussion (please see lines 292-297 in our manuscript).

The bar graphs figure 2 a and 2b are very difficult to read, I would suggest a figure like this, with line and an arrow indicating the introduction of the incentive in the alloHCT intervention group (https://qualitysafety.bmj.com/content/25/5/303)

Response: Thank you for this suggestion. There are only two effective timepoints for allos and two for autos, which there are three timepoints in sequence, but it is not appropriate to make one line connecting the different groups of patients. We believe this format will require many lines to present the same information as one bar plot can present since there are 4-8 time points for each patient depending on their group.

Reviewer #2: This timely article addresses a very hot and important topic of compensation in human microbiome research. The authors performed a prospective study on the impact of financial incentives on stool collections rates among stem cell transplantation patients and demonstrated that a modest incentive of 10 dollars can significantly increase collection rates. This results may inform the design of future studies involving stool collection, given the rapid development of human microbiome research, the value of study is obvious. However, there are still some weaknesses of this paper.

Firstly, the author may need to explain why $10？what's the normal compensation amount in clinical trials setting in their country?

Response: Thank you for this question. We chose $10 because we believed that this amount was modest enough to not unduly influence a patient’s decision to participate in the study (i.e. not excessive enough to be persuasive or coercive) while still serving as an effective incentive to improve patient compliance to stool collection protocols. Furthermore, this modest amount may help other study teams to reproduce this process for their own studies who may have limited resources while still leading to increased collection rates. While it was difficult to find a specific range for the normal compensation amount in the clinical trial setting in the United States , a study from one stool bank paid $40 for stool samples (Chen J, Zaman A, Ramakrishna B, Olesen S. Stool banking for fecal microbiota transplantation: methods and operations at a large stool bank. Frontiers. 2021 April 15; doi: 10.3389/fcimb.2021.622949). Some other studies (cited in the discussion of our manuscript) utilized $5-$20 incentives to try to increase patient compliance.

Secondly, the authors may need to consider the appropriateness of the compensation type regarding the participant population, what about other types of compensation?

Response: We believe that the $10 gas gift card was an appropriate form of compensation for this patient population that consisted of many patients who had to drive very long distances to reach their appointments at the Duke Adult Bone Marrow Transplant Clinic. Thus, we felt that the $10 gas gift card would be an effective incentive without being of too much value to be considered unduly influential. Please also see our response above to reviewer 1’s general comment.

Thirdly, Financial compensation is closely related with undue inducement, which refers to compensation that is so great in amount or nature that it decreases participants' ability to rationally consider participation in the study. The compensation should not be so large that individuals may feel compelled to participate. The authors are suggested to include a discussion on how to avoid undue inducement while incentivise participation and stool collection compliance in research.

Response: Thank you for this suggestion. You bring up a great point, and we agree that research should always be conducted in a manner that promotes the utmost respect for participants and encourages full participant autonomy during the informed consent process. In order to avoid undue inducement while still incentivizing participation and compliance in stool collection research, it is important to choose an amount of compensation that is considered modest enough to not hinder appropriate participant judgment during informed decision making while also ensuring adequate compensation for such a task that is considered quite repulsive by most (i.e. collecting one’s own stool). In order to do this, the amount of compensation should be chosen ahead of time as part of a protocol that is approved by the institutional review board before starting any study related activities. Furthermore, ensuring good research practices by having a study coordinator who is not part of the direct care team carry out a thorough consenting process (e.g. explaining all risks and benefits, emphasizing the decision to participate is completely voluntary, etc.) can help participants to rationally consider participation in the study.

Fourthly, is there any particular vulnerabilities with the patient group the paper focused? would their vulnerabilities affect their participation in fecal collection?

If authors want to get back to the work by addressing these issues, this would make for a far more targeted and clearer discussion.

Response: This is always something very important to consider. There were no particular vulnerabilities with the participants in our study. Please see below for an excerpt about vulnerable populations from our IRB approved protocol of the study: 

“As this study is built around the Duke Adult Bone Marrow Transplant Program, which only treats patients age 18 and older, fetuses, neonates, and children will be excluded. While subjects may be as old as 80 years, adults with cognitive impairment and unable to consent to participation will not be enrolled. Pregnant women are excluded from transplant and will therefore be excluded from this study. Prisoners and institutionalized individuals are also excluded from this study.”

Furthermore, a next step of this project is to investigate how social determinants of health affect stool collection rates in the HCT population in order to see if factors such as socioeconomic status influence compliance and willingness to participate. 

*Copied from PDF comment and already addressed above

the authors may need to consider the appropriateness of the compensation type regarding the participant population, what about other types of compensation? 

Response: Please see above, thank you.

Financial compensation is closely related with undue inducement, which refers to compensation that is so great in amount or nature that it decreases participants' ability to rationally consider participation in the study. The compensation should not be so large that individuals may feel compelled to participate. The authors are suggested to include a discussion on how to avoid undue inducement while incentivise participation and stool collection compliance in research.

Response: Please see above, thank you.

the author may need to explain why $10? and what's the average compensation amount for encouraging compliance in clinical trials in their country? for example, in China, the compensation for each visit in clinical trials setting is between 100-200RMB.

Response: Please see above, thank you.

---

## [Decision Letter · Decision Letter 1]

13 Apr 2022

PONE-D-21-40817R1Financial incentives to increase stool collection rates for microbiome studies in adult bone marrow transplant patientsPLOS ONE

Dear Dr. Sung,

Thank you for submitting your manuscript to PLOS ONE. After careful consideration, we feel that it has merit but does not fully meet PLOS ONE’s publication criteria as it currently stands. Therefore, we invite you to submit a revised version of the manuscript that addresses the points raised during the review process.

Please add the comment to the statistical analysis section as requested by the reviewer.

We look forward to receiving your revised manuscript.

Kind regards,

Peter Gyarmati

Academic Editor

PLOS ONE

Journal Requirements:

Reviewers' comments:

Reviewer's Responses to Questions

**Comments to the Author**

1. If the authors have adequately addressed your comments raised in a previous round of review and you feel that this manuscript is now acceptable for publication, you may indicate that here to bypass the “Comments to the Author” section, enter your conflict of interest statement in the “Confidential to Editor” section, and submit your "Accept" recommendation.

Reviewer #1: (No Response)

Reviewer #2: All comments have been addressed

2. Is the manuscript technically sound, and do the data support the conclusions?

Reviewer #1: Partly

Reviewer #2: Yes

3. Has the statistical analysis been performed appropriately and rigorously? 

Reviewer #1: No

Reviewer #2: Yes

4. Have the authors made all data underlying the findings in their manuscript fully available?

Reviewer #1: Yes

Reviewer #2: Yes

5. Is the manuscript presented in an intelligible fashion and written in standard English?

Reviewer #1: Yes

Reviewer #2: Yes

6. Review Comments to the Author

Reviewer #1: Please add in the statistcial analysis section the sentence written in your answers to reviewers:

'A priori sample size calculation was not performed for this study. While this study was

approved by the Duke IRB and written informed consent was obtained from all study participants,

this study was not formally planned. '

Reviewer #2: The author has addressed issues raised in the comments and greatly improved the quality and soundness of paper, therefore, I recommend the paper to be published.

7. PLOS authors have the option to publish the peer review history of their article (what does this mean?). If published, this will include your full peer review and any attached files.

Reviewer #1: No

Reviewer #2: **Yes: **Yonghui Ma

---

## [Author Response · Author response to Decision Letter 1]

14 Apr 2022

Rebuttal letter: Thank you very much for consideration of our manuscript, and we greatly appreciate your feedback and commitment to the peer review process. Please see our response to the reviewers’ comments. We have revised the paper accordingly.

04-14-2022

PONE-D-21-40817R1

Financial incentives to increase stool collection rates for microbiome studies in adult bone marrow transplant patients

PLOS ONE

Response: the rebuttal letter has been uploaded.

Response: 'Revised Manuscript with Track Changes' has been uploaded.

Response: Manuscript has been uploaded.

Comments to the Author

1. If the authors have adequately addressed your comments raised in a previous round of review and you feel that this manuscript is now acceptable for publication, you may indicate that here to bypass the “Comments to the Author” section, enter your conflict of interest statement in the “Confidential to Editor” section, and submit your "Accept" recommendation.

Reviewer #1: (No Response)

Reviewer #2: All comments have been addressed

2. Is the manuscript technically sound, and do the data support the conclusions?

Reviewer #1: Partly

Reviewer #2: Yes

3. Has the statistical analysis been performed appropriately and rigorously?

Reviewer #1: No

Reviewer #2: Yes

4. Have the authors made all data underlying the findings in their manuscript fully available?

Reviewer #1: Yes

Reviewer #2: Yes

5. Is the manuscript presented in an intelligible fashion and written in standard English?

Reviewer #1: Yes

Reviewer #2: Yes

6. Review Comments to the Author

Reviewer #1: Please add in the statistical analysis section the sentence written in your answers to reviewers:

'A priori sample size calculation was not performed for this study. While this study was

approved by the Duke IRB and written informed consent was obtained from all study participants,

this study was not formally planned. '

Response: The requested information has been added to the statistical analysis of the methods section.

Reviewer #2: The author has addressed issues raised in the comments and greatly improved the quality and soundness of paper, therefore, I recommend the paper to be published.

Response: Thank you.

---

## [Editor Report · Decision Letter 2]

20 Apr 2022

Financial incentives to increase stool collection rates for microbiome studies in adult bone marrow transplant patients

PONE-D-21-40817R2

Dear Dr. Sung,

We’re pleased to inform you that your manuscript has been judged scientifically suitable for publication and will be formally accepted for publication once it meets all outstanding technical requirements.

Kind regards,

Peter Gyarmati

Academic Editor

PLOS ONE
---

## [Editor Report · Acceptance letter]

25 Apr 2022

PONE-D-21-40817R2 

Financial incentives to increase stool collection rates for microbiome studies in adult bone marrow transplant patients 

Dear Dr. Sung:

I'm pleased to inform you that your manuscript has been deemed suitable for publication in PLOS ONE. Congratulations! Your manuscript is now with our production department. 

Kind regards, 

on behalf of

Dr. Peter Gyarmati 

Academic Editor

PLOS ONE